# High prevalence of small intestine bacteria overgrowth and asymptomatic carriage of enteric pathogens in stunted children in Antananarivo, Madagascar

Jean-Marc Collard[1¤a]*, Lova Andrianonimiadana[1], Azimdine Habib[1], Maheninasy Rakotondrainipiana[2], Prisca Andriantsalama[2], Ravaka Randriamparany[2], M. A. N. Rabenandrasana[1], François-Xavier Weill[3], Nathalie Sauvonnet[4], Rindra Vatosoa Randremanana[2], Vincent Guillemot[5], Pascale Vonaesch[4¤b], Philippe J. Sansonetti[4¤a¤c], for the Afribiota Investigators[¶]

**1** Unité de Bactériologie Expérimentale, Institut Pasteur de Madagascar, Antananarivo, Madagascar, **2** Unité d'Epidémiologie et de Recherche Clinique, Institut Pasteur de Madagascar, Antananarivo, Madagascar, **3** Centre National de Référence des *Escherichia coli*, *Shigella* et *Salmonella*, Unité des Bactéries Pathogènes Entériques, Institut Pasteur, Paris, France, **4** Unité de Pathogénie Microbienne Moléculaire, Institut Pasteur, Paris, France, **5** Hub of Bioinformatics and Biostatistics, Institut Pasteur, Paris, France

¤a Current address: The Center for Microbes, Development and Health (CMDH), Institut Pasteur of Shanghai/Chinese Academy of Sciences, Shanghai, China
¤b Current address: Department of Fundamental Microbiology, University of Lausanne, Lausanne, Switzerland
¤c Current Address: Emeritus Professor, Institut Pasteur and Collège de France, Paris, France
¶ The Afribiota Investigators (consortium) are listed in the Acknowledgments
* jean-marc.collard@pasteur.fr

## Abstract

Environmental Enteric Dysfunction (EED) refers to an incompletely defined syndrome of inflammation, reduced absorptive capacity, and reduced barrier function in the small intestine. It is widespread among children and adults in low- and middle-income countries and is also associated with poor sanitation and certain gut infections possibly resulting in an abnormal gut microbiota, small intestinal bacterial overgrowth (SIBO) and stunting.

We investigated bacterial pathogen exposure in stunted and non-stunted children in Antananarivo, Madagascar by collecting fecal samples from 464 children (96 severely stunted, 104 moderately stunted and 264 non-stunted) and the prevalence of SIBO in 109 duodenal aspirates from stunted children (61 from severely stunted and 48 from moderately stunted children). SIBO assessed by both aerobic and anaerobic plating techniques was very high: 85.3% when selecting a threshold of $\geq 10^5$ CFU/ml of bacteria in the upper intestinal aspirates. Moreover, 58.7% of the children showed more than $10^6$ bacteria/ml in these aspirates. The most prevalent cultivated genera recovered were *Streptococcus*, *Neisseria*, *Staphylococcus*, *Rothia*, *Haemophilus*, *Pantoea* and *Branhamella*. Feces screening by qPCR showed a high prevalence of bacterial enteropathogens, especially those categorized as being enteroinvasive or causing mucosal disruption, such as *Shigella* spp., enterotoxigenic *Escherichia coli*, enteropathogenic *E. coli* and enteroaggregative *E. coli*. These pathogens were detected at a similar rate in stunted children and controls, all showing no sign of

**Data Availability Statement:** All relevant data used for the analyses are within the manuscript and Supporting information files.

**Funding:** This project was funded by the Total and Petram Foundations and by Institut Pasteur. LA, PA and RR wages were supported by the Total Foundation and MR wage by the Petram Foundation. PV was supported by an Early Postdoctoral Fellowship (P2EZP3_152159), an Advanced Postdoctoral Fellowship (P300PA_177876) as well as a Return Grant (P3P3PA_17877) from the Swiss National Science Foundation, a Roux-Cantarini Fellowship (2016), a L'Oréal-UNESCO for Women in Science France Fellowship (2017) and an Excellence Scholarship from the University of Basel (Forschungsfonds, 2019). The funders had no role in study design, data collection and analysis, decision to publish, or preparation of the manuscript.

**Competing interests:** The authors have declared that no competing interests exist.

severe diarrhea the day of inclusion but both living in a highly contaminated environment (slum-dwelling). Interestingly *Shigella* spp. was the most prevalent enteropathogen found in this study (83.3%) without overrepresentation in stunted children.

## Author summary

About 2 million children under the age of 5 suffer from stunted growth in Madagascar. Although deficient diet is the major cause of undernutrition, impaired absorption or assimilation caused by Environmental Enteric dysfunction (EED) has been proposed to play an important role in stunting. EED is widespread among children and adults in low- and middle-income countries (LMIC) and is also associated with undernutrition, poor sanitation, certain gut infections resulting in an abnormal gut microbiota and small intestinal bacterial overgrowth (SIBO) although the role of SIBO in EED remains unclear. The current study highlights the presence at high concentrations of bacterial taxa usually found in the oro-pharyngeal sphere in a high number of duodenal fluids of stunted children. This uncommon presence suggests a decompartmentalization of the gastrointestinal tract and a possible pro-inflammatory effect due to the ectopic presence of some of these bacteria in the duodenum.

The study also points to a high prevalence of enteropathogens (especially *Shigella* spp.) in the feces of both stunted and control children, hence preventing from proposing a direct association with stunting. This suggests that, beside combatting poverty and improving diet, environmental sanitation, quality of water sources, hygiene promotion and health education are key points to mitigate stunting and restore nutritional benefits.

## Introduction

Stunting is a major public health and economic development concern in Madagascar which is the 5th most affected country in the world with 48.5% of all children under 5 (ca. 2 million) suffering from stunted growth [1]. Although deficient diet is the major cause of undernutrition, other factors such as Environmental Enteric Dysfunction (EED) has been proposed to be an associate driver of stunting in low and middle-income countries (LMICs) [2,3]. EED is a chronic inflammatory condition of the gut occurring among children living in unsanitary conditions [4], or among adults returning from deployment to LMICs [5,6]. According to estimates, greater than 75% of all children in LMICs could suffer from this syndrome at different degrees of severity [7]. Certain gut infections resulting in an abnormal gut microbiota and small intestinal bacterial overgrowth (SIBO) may also play a role in EED [5].

A study in Bangladesh found enteric infections in the first two years of life, mainly *Shigella* and Enterotoxigenic *E. coli* (ETEC), to be associated with EED and stunting [8]. Two additional studies found entero-aggregative *E. coli* (EAEC) and *Campylobacter* to be associated with markers of gut inflammation and stunting [9,10]. Even in the absence of diarrhea, these recurrent enteric infections may result in an imbalanced gut microbiota (i.e. dysbiosis) leading to growth faltering mediated through systemic inflammation [11].

SIBO is defined as the presence of excessive bacteria (greater than $10^5$ CFU/ml) in the small intestine. It is frequently implicated as the cause of chronic diarrhea and malabsorption. SIBO can be measured noninvasively by hydrogen breath testing or by anaerobic and aerobic cultures of endoscopically aspirated upper gastrointestinal fluid [12]. In a study performed on

Bangladeshi children, SIBO was associated with a decreased length-for-age Z score since birth and poor sanitation but was independent of frequent or recent diarrheal disease. It was also associated with intestinal inflammation but not with increased permeability or systemic inflammation [13].

Both repetitive gut infections and SIBO can co-exist in an intestine weakened by undernutrition and might lead themselves to micronutrient malabsorption fueling a vicious cycle with long-lasting effects.

AFRIBIOTA is a case-controlled study for stunting, conducted in children in Antananarivo, Madagascar and in Bangui, Central African Republic, in which 460 children aged 2–5 years with no overt signs of gastrointestinal disease were recruited in each country [14]. In a preliminary study (feces from 153 children and 12 duodenal aspirates) [15], we showed vast majority of the stunted children (>80%) showed SIBO dominated by bacteria normally residing in the oro-pharyngeal tract. There was also an overrepresentation of oral bacteria in fecal samples of stunted children. *Escherichia coli/Shigella* spp. and *Campylobacter* spp. were found by a metataxonomic approach to be more prevalent in stunted children, while the butyrate-producers (*Clostridia*) were reduced [15].

The current research aimed to assess SIBO and gut infections in a population of children (N = 464) recruited in Antananarivo, Madagascar, using a quantitative culture-based technique on duodenal aspirates from stunted children and a more sensitive and accurate molecular-based approach (qPCR) than the classical metataxonomic approach [16,17,18], targeting the main enteropathogenic bacterial species and pathobionts in the feces.

## Methods

### Ethics statement

The study protocol for AFRIBIOTA has been approved by the Institutional Review Board of the Institut Pasteur (2016–06/IRB) and the National Ethical Review Boards of Madagascar (55/MSANP/CE, May 19th 2015). All participants received oral and written information about the study and the legal representatives of the children provided written consent to participate in the study.

### Recruitment of participants

The case control was extensively described in a preliminary study [15] and is similar to the full-study. Briefly, the study population comprises HIV-negative children aged 2 to 5 years, neither suffering from acute malnutrition, nor from any other severe diseases (such as dysenteric syndrome, severe acute respiratory infections (SARI)/influenza-like illness (ILI), meningitis, malaria, acute otitis media, varicella, measles, . . .) who were recruited in the community of the two districts of Ankasina or Andranomanalina Isotry, and in two hospitals (Centre Hospitalier Universitaire Joseph Ravoahangy Andrianavalona (CHU-JRA) and Centre de Santé Materno-Infantile, Tsaralalana) in Antananarivo, Madagascar. Children were recruited either in the community (community-recruited children) or directly in the hospital (hospital-recruited children). The caregivers of all children received oral and written explanations of the study and provided written consent to participate. Anthropometric measurements were collected at the community-health centers and in the hospital. All children were admitted to the hospital for sample collection. Height was measured to the nearest 0.1 cm in a standing position using collapsible height boards; weight was measured using a commercial weighing scale to the nearest 100 g. All measurements were taken at least twice and repeated if the measurements were more than 0.1 cm/100 g apart from each other. The children were classified according to the median height of the WHO reference population [16] in three different

groups: severe stunting (height-for-age z-score $\leq$ -3SD), moderate stunting (height-for-age z-score between -3SD and -2SD) and not stunted (height-for-age z-score $\geq$ -2SD). The non-stunted (control subjects) were matched for living area and sampling time period and were recruited during the entire study period (December 2016—March 2018). The realized sample size was 490 children: 211 stunted children and 279 not stunted. All children (hospital- and community-recruited children) were asked to come to the hospital for sample collection for the safety of the children (see hereafter).

## Collection of stools, gastric and duodenal aspirates

All those conditions have been previously described in Vonaesch et al., 2018 [15]. Briefly, stools were collected in the morning at the hospital (directly before coming to the hospital for the community-recruited children) and the time of defecation recorded. If community-recruited children were able to again emit feces in the hospital, these feces were also collected and directly snap-frozen in liquid nitrogen before being transferred to -80˚C. The time spent between defecation and freezing (liquid nitrogen/-80˚C) was recorded on specific tracking sheets for each sample. The time for freezing the emitted feces was comprised between 6 min and 23.66h with a mean of 3.69h.

Gastric and duodenal samples were collected using a pediatric nasogastric tube with stent (Vygon, France), and were only collected for stunted children due to ethical concerns. The nasogastric tube was introduced in the infant's nose sitting in an upright position and moved to the stomach. Proper placement in the stomach was controlled using the syringe-air test and confirmed by an acid pH (pH $\leq$ 4). Once at least 4 ml of liquid were aspirated, the tube was moved 5 cm forward. The child was then laid down on the right side as to facilitate the tube's passage in the duodenum. Pyloric passage was monitored regularly and was confirmed when the pH reached a pH $\geq$ 5. In this case, a second aspiration was performed and identified as duodenal. The procedure was interrupted if the pH did not change to a pH $\geq$ 5 within 1h30 min after introducing the tube. We also ensured that the first ml of aspiration from both the stomach and the duodenum were discarded. This allowed flushing out possible contaminating bacteria and minimize the carry-over from the more proximal compartments. 100 µl of fresh duodenal sample were inoculated directly in 0.9 ml of Robertson's Cooked Meat (RCM) medium and processed for culture (see below). The rest of the aliquots were directly snap-frozen in liquid nitrogen and then transferred the same day to a -80˚C freezer.

## Culture of duodenal aspirates, feces and identification of colonies

RCM-diluted duodenal aspirations were brought within a time frame of 30 min to the Institut Pasteur of Madagascar, where they were further diluted and streaked on plates according to the protocol described in Chandra *et al.* [17]. In brief, the duodenal samples were analyzed using plating techniques in aerobic and anaerobic culture conditions after the RCM-diluted duodenal aspirates were serially diluted (1:100, 1:1,000) in phosphate buffered saline containing 1% peptone and 20 µL of each dilution and inoculated onto 3 separate culture plates. Aerobic culture was performed on chocolate agar, 5% sheep blood agar, and MacConkey's medium. Anaerobic cultures were performed directly from the RCM tube (without dilution) onto 5% sheep blood agar containing haemin (5 µg/ml) and vitamin K (menadione) (1 µg/mL). The plates were incubated in an anaerobic jar or aerobically at 37˚C for 48 h. Total bacterial count per ml of duodenal aspirate was calculated based on the total count on the chocolate agar plates (incubated aerobically) and the anaerobic culture. The following formula was used to calculate the total CFU load: No. of CFU per 20 µL of fluid inoculated x $\frac{1}{20}$ x 1,000 X dilution on plate.

The isolated colonies were counted by morphotype and at the beginning 2–3 colonies for every morphotype were re-isolated. Re-isolated colonies were all identified by MALDI-TOF mass spectrometry (Bruker Biotyper, Bruker Daltonics, Bremen, Germany). As colonies from the same morphotype from the same plate were all identical (same identification), finally only one colony/morphotype was isolated and subcultured. Cultures were considered positive for SIBO if the total bacterial count was $\geq 10^5$ CFU per mL of duodenal fluid [12,17].

To isolate *Shigella* spp. from the last 143 feces samples, a fresh scoop of feces taken with an inoculation loop was suspended in 5ml of physiological water and 20 μL were plated on Hektoen and XLD media. Suspected *Shigella* isolates (green or pink colonies on Hektoen or XLD, respectively) were purified, stored and identified with API galleries. There were sent to the French National Reference Center for *E. coli*, *Shigella* and *Salmonella* (FNRC-ESS), Institut Pasteur, Paris for species confirmation and serotyping. Serotyping was done by slide agglutination assays using a complete set of antisera allowing recognition of all described *Shigella* serotypes [18].

## Data collection

To assess risk factors for acquiring pathogens, a questionnaire was developed and administered to children and their caregivers. In brief, the questionnaire contained four sections: 1. Socio-demographic data: age, gender, community setting, education and occupation of parents, family marital status; 2. Environmental factors: housing conditions (proximity of the housing to landfill, soil type, lavatories and showers) and quality of drinking water; 3. Behavior habits: type of toilet commonly used, hand washing habit, exposure to sewage and garbage; 4. Medical status/history: dental cavities, cough, stuffy nose, runny nose, delivery mode.

## Statistical analysis

The data were encoded in an Excel (microbiological data) and an Access (socioeconomic data) database and analyzed using the R statistical software version 3.6.2. Comparisons between groups (controls and stunted children MS+SS) were determined using Pearson's χ2-test or Fisher's exact test, as appropriate. Values of $p < 0.05$ were considered to be statistically significant.

For risk factors analysis, bivariate logistic regression analysis was carried out between factors (S1 and S2 Tables) and the presence of SIBO using the Chi-squared test or the Fisher's exact test. Variables having P-value of $\leq 0.25$ were entered into multivariate logistic regression for final analysis. Additionally, we correlated the pathogen presence in stools with the measure of CFU/ml grown from duodenal aspirates in aerobic and anaerobic culture conditions using the non-parametric Wilcoxon's rank sum test.

For the clustering analysis, the data consisted in 0s and 1s corresponding to the infection status (detection of the pathogen by qPCR) of the children: 0 = uninfected and 1 = infected. The infection profiles for the children obtained were clustered with a Hierarchical Clustering based on a binary distance coupled with Ward's agglomeration method. No clustering was applied on the variables.

## DNA extraction and real-time PCR

Samples were extracted by commercial kits using a Qiacube instrument (Cador Pathogen 96 QIAcube HT Kit, Qiagen France SAS, Courtaboeuf, France). DNA extractions were performed following the manufacturer's recommendations with an additional bead-beating step to increase mechanical disruption. In brief, 200 mg of freshly thawed sample were mixed with 1.4 mL of ASL buffer at 4˚C and vigorously vortexed for 1 min. The suspension was transferred

into a Pathogen Lysis Tube S (Qiagen) containing 2 mg of sterile glass beads (100μM diameter). Samples were mechanically disrupted using a TissueLyser II (Qiagen Retsch GmbH, Hilden, Germany) for 10 min at 30Hz. The suspension was then incubated at 95˚C for 5 min, vortexed for 15 sec and centrifuged at 14,000 x *g* for 1 min to ensure that no solid particles were transferred to the subsequent steps. A volume of 1.2 mL of the supernatant was then transferred into a new tube containing an InhibitEX tablet (Qiagen). Samples were vigorously vortexed for at least 1 min or until complete dissolution of the InhibitEX tablet. Three steps of centrifugation and supernatant transfer were performed thereafter (14,000 x *g* for 3 min). The remainder of the protocol was performed as recommended by the manufacturer. All samples were eluted in 150 μL AE buffer. DNA concentrations and purity were assessed via 260/280 and 260/230 absorbance ratios by spectrophotometry (Nanodrop 2000 Spectrophotometer, Thermo Fisher Scientific, Waltham, MA, USA). Samples were stored at -80˚C until molecular analyses.

Amplifications were carried out in an ABI StepOne instrument (Applied Biosystems, Nairobi, Kenya). After an initial denaturation step (95˚C for 10 min), 45 cycles of two-step PCR (95˚C for 15 s and 50–56˚C for 60 s according to the duplex reaction) were performed in 5 parallel duplex reactions, targeting a broad range of diarrheagenic bacterial agents as described in Table 1. The result for each agent was recorded as the *Ct* value, which is inversely related to the pathogen load in each specimen. *Ct* values < or equal to 37 were considered as positive. Standard curves for each target were established and the linearity ensured the *Ct* cutoff value of 37 was applicable.

**Bacterial agents and target sequences.**   The targets for real-time PCR are presented in Table 1. Bacterial PCRs were developed from available publications concerning suitable target regions. All sequences were controlled and blasted in GenBank and some primers were adapted according to new retrieved sequences from GenBank. *Salmonella* spp. and *Shigella* spp. were identified by amplification of the outer-membrane protein C and the invasion plasmid antigen H (*ipaH*) gene (which also may be present in enteroinvasive *E. coli* [EIEC]), respectively.

The presence of *ipaH* gene present on the virulence plasmid and on the chromosome was also tested by conventional PCR with the primers described in Phantouamath *et al*. [23]

For enterotoxigenic *Escherichia coli* (ETEC), heat-labile toxin (*eltB*) and heat-stable toxin (*estA*) coding regions were targeted. For enteropathogenic *E. coli* (EPEC), the bundle-forming pilus (encoded by the *bfpA* gene) carried by the EPEC adherence factor (EAF) plasmid and the intimin (*eae* gene for EPEC attaching and effacing), an outer membrane adhesion essential for the intimate attachment of the EPEC or enterohemorrhagic *E. coli* (EHEC) to enterocytes were the targets. For enteroaggregative *E. coli* (EAEC), *aggR* and *aaiC* genes were amplified. The *aggR* gene encodes a transcriptional activator of the aggregative adherence fimbriae expression in enteroaggregative *E. coli* and *aaiC*, is part of the *aai* gene cluster, encoding a type VI secretion system. When one of the two targets was detected for ETEC (*estIa* or *eltB*) or EAEC (*aggR* or *aaiC*), the pathogen was considered to be present. Since *eae* can be present both in EPEC and EHEC, we considered only the presence of *bfpA* gene for EHEC detection.

Fibronectin-binding protein (*cadF*) gene and the cholera toxin (CT) subunit A gene (*ctxA*) were the targeted genes for *Campylobacter jejuni/coli* and *Vibrio cholerae*, respectively. Sufficient amplification efficiencies were documented for each realtime PCR by analyzing serial dilutions of pUC57 plasmids carrying all synthetic target inserts (GeneCust Europe, Dudelange, Luxembourg). By comparing *Ct* values for each target amplified alone or in duplex reactions, it was confirmed that performance was not compromised by multiplexing (5 duplex reactions). In addition to optimization, which focused on analytical sensitivity, diagnostic accuracy was evaluated by analyzing well-characterized bacterial strains from the Culture

**Table 1. Primers and probes targeting DNA of diarrheagenic agents.**

| Pathogen | Duplex/ annealing t° | Forward primer | Reverse primer | Probe | Fluorophores | Target gene/ Access number –GenBank | Reference |
|---|---|---|---|---|---|---|---|
| *Salmonella* spp. | 1 / 50°C | CGGGTTGCGTTATAGGTCTGA | TGAAATACGATGCGAACAACATC | AATACTGCGCTGCCAGAT | HEX-BQ1 | Outer membrane protein, *ompC* / AIH09487.1 | [19] |
| *Shigella* spp. | 2 / 56°C | ACCGGCGGCTCTGCTCTC | GCAATGTCCTCCAGAATTTCG | CTGGGCAGGGAAATGTTCCGCC | HEX-BQ1 | invasion plasmid antigen H, *ipaH* / DQ132807.1 | [19] |
| ETEC | 1 / 50°C | AAGCATGAATAGTAGCAATTACTGCT -> AAGCATGAATRGTAGCAATTACTGCT* | TTAATAGGCACCCGGTACAAGCA | AACAACACAATTCAC -> TACAACACAATTCAC* | FAM-BQ1 | Heat-stable enterotoxin ST, *estIa* / M29255.1 | [19]* |
| ETEC | 2 / 56°C | TCCGGCAGAGGATGGTTACA | CCAGGGTTCTTCTCTCCAAGC | AGCAGGTTTCCCACCGGATCACC | FAM-BQ1 | Heat-labile enterotoxin LT, *eltB* / BAI49232.1 | [19] |
| EPEC | 3 / 50°C | CATTGATCAGGATTTTTCTGGTGATA | CTCATGCGGAAATAGCCGTTA | ATACTGGCGGAGACTATTTCAA | FAM-BQ1 | Intimin, *eae* / CAG17538.1 | [20] |
| EPEC | 3 / 50°C | TGGTGCTTGCGCCTTGCT | CGTTGCGCTCATTACTTCTG | CAGTCTGCGTCTGATTCCAA | HEX-BQ1 | bundle-forming pilus, *bfpA* / AB247927.1 | [20] |
| EAEC | 4 / 52°C | GAATCGTCAGCATCAGCTACA | CCTAAAGGATGCCCTGATGA | CGGACAACTGCAAGCATCTA | FAM-BQ1 | transcriptional activator of aggregative adherence fimbriae expression, *aggR* / AF411067.1 | [21] |
| EAEC | 4 / 52°C | CATTTCACGCTTTTTCAGGAAT | CCTGATTTTAGTTGATTCCCTACG | CACATACAAGACCTTCTGGAGAA | HEX-BQ1 | part of the *aai* gene cluster, encoding a type VI secretion system, *aaiC* / FN554766.1 | [21] |
| *Campylobacter jejuni* -> *Campylobacter jejuni/coli* | 5 / 53°C | CTGCTAAACCATAGAAATAAAATTTCTCAC -> CWGCTAAACCATARAAATAAAATTTCTCAC* | CTTTTGAAGGTAATTTAGATATGGATAATCG -> YTTTGAAGGTAATTTAGATATGGATAATCG* | CATTTTGACGATTTTTGGCTTGA -> CATTTTGAYGATTTTTGGCTTGA* | HEX-BQ1 | Fibronectin-binding protein, *cadF* / AJK71638.1 | [22]* |
| *Vibrio cholerae* | 5 / 53°C | CCACTTAGTGGGTCAAACTATATTGTC | ATGCCCCTAATACATCATTAACGTT | AGCCACTGCACCCAA | FAM-BQ1 | Cholera toxin A subunit, *ctxA* / X58785.1 | [19] |

* with modifications in the nucleotide sequence of primers and/or probe

Collection, Pasteur Institute of Madagascar (*Salmonella* Typhimurium [BEX125], *Shigella flexneri* [BEX126], *E. coli* Diego 1120 ETEC (*astA*, *estA*, *eltB*, *uidA*) [BEX228], *E. coli* Tul 2322 EPEC (*escV*, *bfpA*, *uidA*, *eae*) [BEX217], *E. coli* Morondava 1612 EAEC (*eae*, *astA*, *aggR*, *pic*, *aaiC*, *uidA*) [BEX216], *Campylobacter jejuni* [BEX112], *Vibrio cholerae* [BEX227]). Negative and positive controls were included in each qPCR amplification run.

## DNA extraction, whole genome sequencing and phylogenetic analysis

DNA was extracted from the *Shigella* isolates with cador Pathogen 96 QIAcube HT Extraction Kit (QIAGEN, Paris, France) on a Qiacube HT from 5 mL of liquid cultures grown overnight at 37˚C in Luria-Bertani infusion medium, following the manufacturer's protocol. Purity and DNA quantity were assessed using Nanodrop spectrophotometer (Thermo Fisher Scientific, Waltham, MA, USA). As previously described [24], illumina sequencing libraries were prepared by using Nextera XT DNA Sample Kit (Illumina, San Diego, CA, USA) with indexed-encoded adapters from Illumina, according to the manufacturer's instructions. The libraries were pooled for sequencing on NextSeq 500 platform (Illumina) using $2 \times 150$-bp runs. FqCleaner (version 3.0) was used to eliminate adaptor sequences, reduce redundant or over-represented reads, correct sequencing errors, merge overlapping paired reads, and discard reads with Phred scores (measure of the quality of identification of nucleobases generated by automated DNA sequencing) <20. The Illumina sequence data were assembled using Spades software [25].

The average nucleotide identity (ANI) values were calculated in EzGenome (https://www.ezbiocloud.net/taxonomy) [26] and Genome-to-Genome Distance Calculator (GGDC; http://ggdc.dsmz.de) [27], respectively. Whole-genome-based taxonomic analysis was performed by Type Strain Genome Server (TYGS) (at https://tygs.dsmz.de) [28]. The phylogenomic tree was constructed using FastME [29] from the genome blast distance phylogeny (GBDP). The trees were rooted at the midpoint [30]. Branch supports were inferred from 100 pseudo-bootstrap replicates. MLST, wgMLST, rMLST and cgMLST were obtained using Enterobase and virulence factor were identified using abricate package.

This whole-genome shotgun project has been deposited at DDBJ/ENA/GenBank under the accession number JAIQVZ000000000 for *Shigella flexneri* HJRA178 and JAIQWA000000000 for *Shigella flexneri* HJRA198. The version described in this paper is version JAIQVZ010000000 and JAIQWA010000000. Raw sequence data for those strains were deposited under strain ESC_WA 5556AA for *Shigella flexneri* HJRA198 and ESC_WA5569AA for *Shigella flexneri* HJRA178 in Enterobase.

## Gentamicin invasion assay

The gentamicin invasion assay was performed to determine the invasion rate of viable *Shigella* isolates inside the HEp-2 cells. The methodology used was described in the Bio-protocol 9(13): e3292. [31]. In brief, the amount of $3.10^5$ HEp-2 cells per well were initially seeded in a 6-well microplate. After 24 hours of incubation at 37˚C in 10% $CO_2$, the cell cultures were inoculated with *Shigella* isolates at multiplicity of infection (MOI) for this experiment of 5 (about $5 \times 10^6$ bacteria). The infected cells were incubated for 1 hour, washed three times with PBS and incubated for an additional hour with the gentamicin solution. The antibiotic solution was removed, the cells were washed three times with PBS and the infected cells were lysed with 1 ml of sodium deoxycholate solution. The remaining *Shigella* were quantified by CCU methodology and performed in duplicate. These CCU were compared with the initial values of *Shigella* suspensions.

## Results

### Description of the study population

The study population of the Afribiota full-project (Madagascar) comprises 490 children recruited in the community of the two districts of Antananarivo, Madagascar, of which 464 were able to provide enough feces for molecular analysis. Among these 464 children, 96 were severely stunted -SS-, 104 moderately stunted -MS- and 264 non-stunted considered as controls—C -. Sex ratio (M/F) was distributed as follows: 0.92, 1.36 and 1.2 for SS, MS and C, respectively. Duodenal aspirates (N = 109) were analyzed using bacteriological techniques and stools were analyzed by qPCR (Fig 1).

### SIBO

In our study, a total of 165 duodenal aspirates were collected; however, due to several limitations and constraints (i.e. pH $\geq$ 5, enough volume for subsequent analyses), only 109 samples were investigated, 61 from SS and 48 from MS children (Fig 1). This is a substantial extension of the initial dataset presented in ref. 15 (12 duodenal samples). The duodenal samples were analyzed using plating techniques in aerobic and anaerobic culture conditions at 37˚C. The prevalence of SIBO ($\geq 10^5$ CFU/ml of bacteria in the upper intestinal aspirates) was 85.3% (93 children out of 109) with 58.7% hosting more than $10^6$ bacteria/ml (Table 2). The gender distributions were similar for the two groups, with a sex ratio of 0.94 and 0.63 for SS and MS, respectively. The mean age was 41.3 and 40.1 months for SS and MS, respectively. Per aspirate, two to eleven different morphotypes were randomly chosen and identified. In total, 51 different species were identified. *Streptococcus*, *Neisseria*, *Staphylococcus*, *Rothia*, *Haemophilus*, *Pantoea* and *Branhamella* were the most prevalent genera cultivated (Fig 2).

### SIBO and risk factors

SIBO was determined in stunted children only. Consequently, risk factors were assessed only for these children. According to the univariate analysis, age group, dental cavities, mode of delivery, lavatories, showers, household waste and soap usage for hand washing (mother) showed a p-value less than 0.25 to SIBO. A multivariable logistic regression analysis was carried out on these variables and none of these were found to be statistically associated with SIBO (p>0.05) (Tables 3 and S1 and S2). Regarding the pH values measured in the stomach samples from stunted children, they were low and there were no significant differences in a bivariate analysis for stomach pH and SIBO (S3 Table).

### Pathogens detection rates

In total, 464 fecal samples (264 controls -C-, 104 MS and 96 SS children) were included in the analysis. Negative and positive (pUC57 plasmids carrying synthetic target inserts and reference strains) controls were included in each qPCR amplification run. Table 4 (see also S4 Table) presented the percentages of bacterial diarrheagenic agents found in stunted children (moderately and severely stunted) and controls ($C_T$ values < or equal to 37.0 were considered as positive). At least one bacterial pathogen was detected for 91.8% (90.9% in controls (N = 240) and 92% in stunted children (99 MS and 85 SS)).

All pathogens were detected at similar prevalence in MS/SS and in C (no statistical differences). The highest prevalence was found for the *ipaH* gene (*Shigella* spp. or enteroinvasive *E. coli*) with values up to 90.4% in MS and equal to 83.3 for the three groups altogether (MS, SS and C). The mean, median, lowest and highest $C_T$ values for *ipaH* were 30, 31.5, 14.5 and 45, respectively. This suggests that the bacterial load was relatively high (Fig 2). The $C_T$ values

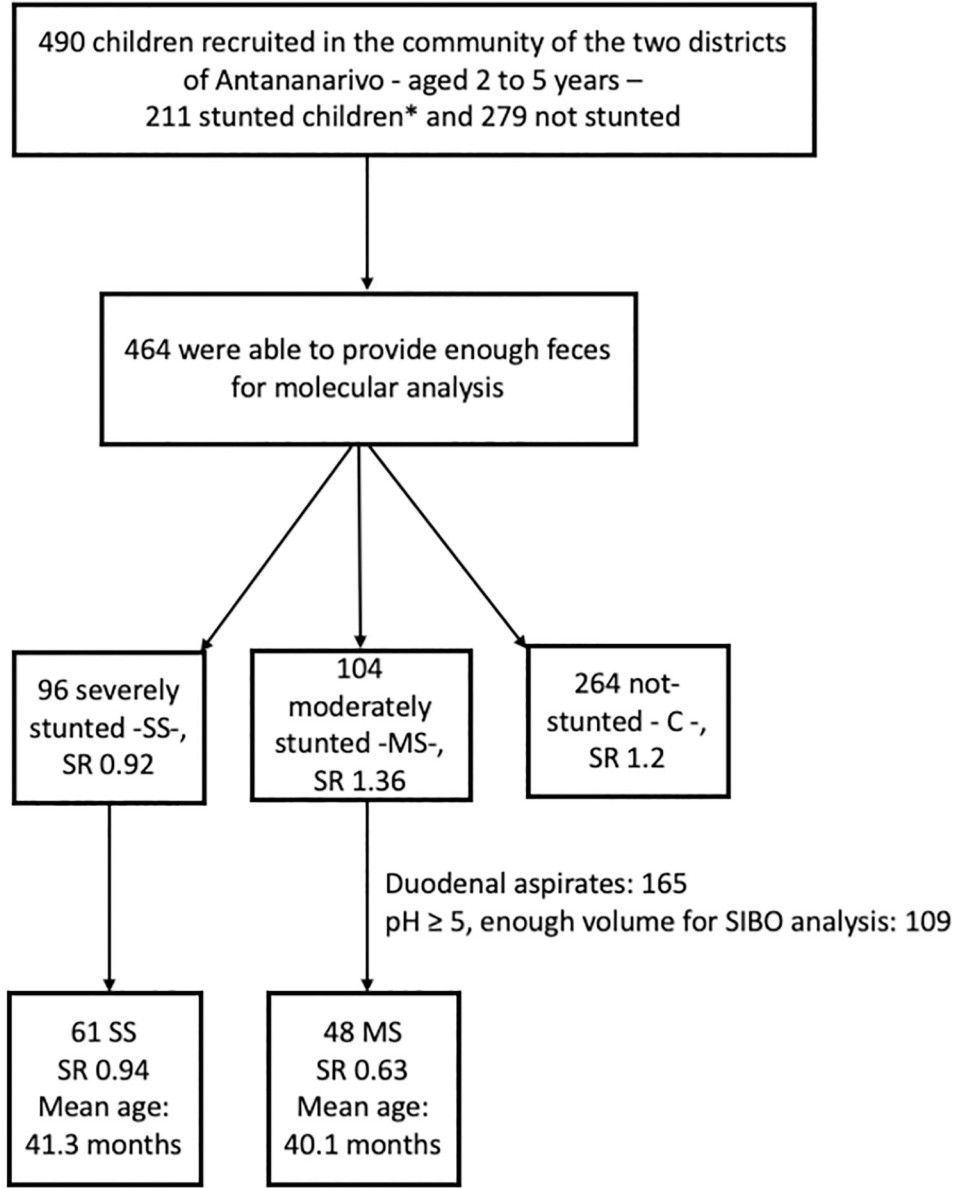

**Fig 1. Flowchart inclusion process.** HIV-negative children aged 2 to 5 years, neither suffering from acute malnutrition, nor from any other severe disease recruited in the community of the two districts of Ankasina or Andranomanalina Isotry, and in two hospitals (Centre Hospitalier Universitaire Joseph Ravoahangy Andrianavalona (CHU-JRA) and Centre de Santé Materno-Infantile, Tsaralalana) in Antananarivo, Madagascar. They were classified in three different groups*: severe stunting—SS—(height-for-age z-score $\leq$ -3SD), moderate stunting—MS—(height-for-age z-score between -3SD and -2SD) and not stunted—C for controls—(height-for-age z-score $\geq$ -2SD). SR: Sex ratio.

were not significantly lower in stunted children than controls for *ipaH*, as for the other genes/pathogens (Fig 3).

When considering the presence of only one of the two genes for ETEC and EAEC, and only *bfpA* for EPEC since *eae* can be present both in EPEC and EHEC, ETEC and EAEC were present in about one-third (ETEC = 32.8 and EAEC = 29.9%) of the children (C, MS and SS children) and EPEC in 14% of the children. *Campylobacter jejuni/coli* and *Samonella* spp. were

**Table 2. CFU values in duodenal aspirates of stunted children (N = 109).** SIBO is defined as greater than $10^5$ CFU/ml of upper intestinal aspirate as assessed by both anaerobic and aerobic cultures [16].

| CFU values | N (%) |
| --- | --- |
| 0 | 7/109 (6.4) |
| $10^2 < cfu < 10^5$ | 9/109 (8.3) |
| $10^5 < cfu < 10^6$ | 29/109 (26.6) |
| $10^6 < cfu < 10^7$ | 29/109 (26.6) |
| $10^7 < cfu < 10^8$ | 28/109 (25.7) |
| $cfu >= 10^8$ | 7/109 (6.4) |

present in 13.6% and 8.2% of all children. No *Vibrio cholerae* were found. Infection with more than one pathogen was a common finding for both stunted children and asymptomatic controls: Out of 464 children, 173 (37.3%) carried 3 or more (until 5) pathogens, 151 (32.5%) and 107 (23%) carried two or one pathogen, respectively. No pathogens were detected in only 7.1% (N = 33) of the children.

Using the Wilcoxon's rank sum test, we correlated the pathogen presence in stools with the CFU/ml values from duodenal aspirates in aerobic and anaerobic culture conditions. The only pathogen significantly associated with the duodenal CFU count was *Campylobacter* spp. (p = 0.026) (S5 Table).

The distributions of the CFU in duodenal aspirates with SIBO matching with feces contaminated or not by *Campylobacter* spp. are presented in S1 Fig.

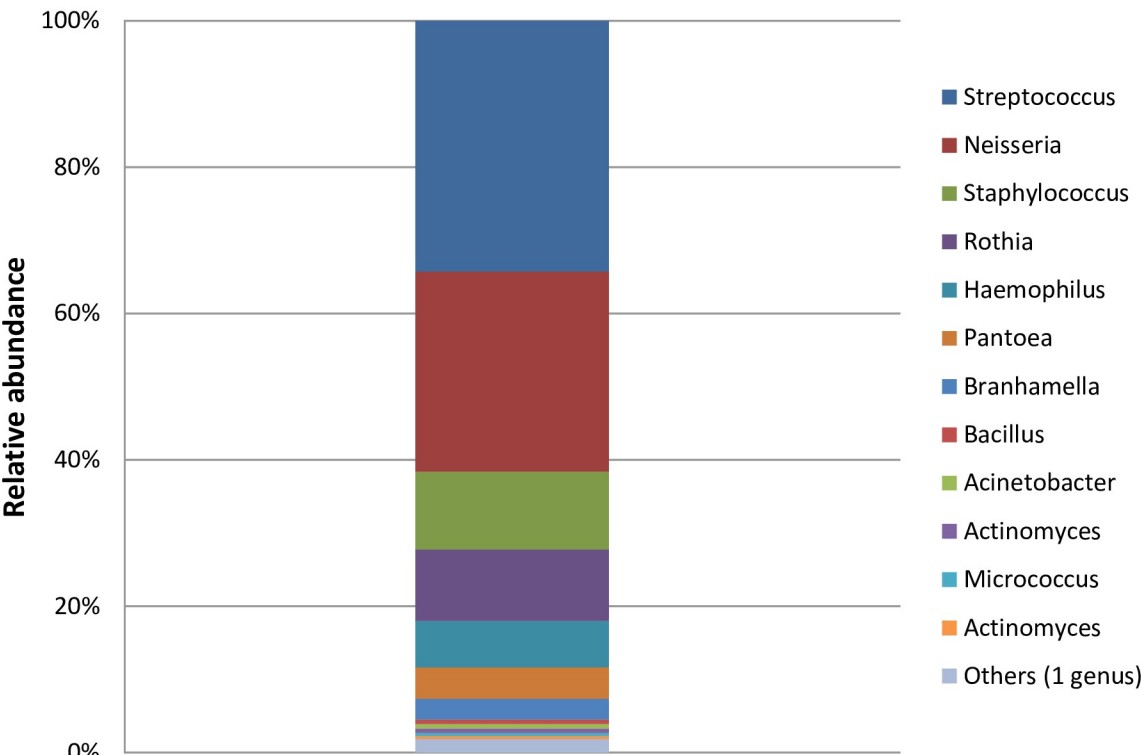

**Fig 2. Number of isolates of the 12 most-abundant genera identified by culture techniques in duodenal aspirates of stunted children.** The color code for the different genera is given on the right.

**Table 3. Possible risk factors associated with SIBO.** Only variables having a P-value ≤0.25 in the univariate analysis (S2 Table) were entered into multivariate logistic regression for a final analysis (S3 Table). All variable were recorded by field workers (clinical research associates).

| Variables | SIBO | | p-values |
|---|---|---|---|
| | positive | negative | |
| **Age group** | | | |
| [2–3] | 46 | 6 | 0.13 |
| [4–5] | 43 | 14 | |
| **Dental cavities** | | | |
| yes | 34 | 11 | 0.25 |
| no | 55 | 9 | |
| **Delivery mode** | | | |
| Vaginal delivery | 80 | 20 | 0.20 |
| Caesarean delivery | 9 | 0 | |
| **Lavatories** | | | |
| Collective | 79 | 17 | 0.14 |
| Individual | 6 | 0 | |
| No toilets | 4 | 3 | |
| **Showers** | | | |
| Inside house | 2 | 2 | 0.14 |
| Outside house | 44 | 7 | |
| No showers | 43 | 11 | |
| **Household waste** | | | |
| Burn | 13 | 0 | 0.12 |
| Threw | 76 | 20 | |
| **Soap usage for hand washing (mother)** | | | |
| Before meals | 50 | 15 | 0.07 |
| Before and after meals | 38 | 4 | |
| After meals | 1 | 0 | |
| Never | 0 | 1 | |

**Table 4. Comparison between stunted children and controls for the presence of bacterial diarrheagenic agents in fecal samples.**

| | % for controls (N = 264) | % for MS = (N = 104) | % for SS = (N = 96) | P-value* |
|---|---|---|---|---|
| *Salmonella* spp. | 7.95 (N = 21) | 5.21 (N = 5) | 12.5 (N = 12) | 0.13 |
| *Shigella* spp. | 80.68 (N = 213) | 90.38 (N = 94) | 84.38 (N = 81) | 0.07 |
| ETEC *estIa* | 4.17 (N = 11) | 5.77 (N = 6) | 6.25 (N = 6) | 0.66 |
| ETEC *eltB* | 29.92 (N = 79) | 40.38 (N = 42) | 32.29 (N = 31) | 0.15 |
| EPEC *bfpA* | 13.26 (N = 35) | 11.54 (N = 12) | 18.75 (N = 18) | 0.29 |
| EPEC *eae* | 37.50 (N = 99) | 36.54 (N = 38) | 36.46 (N = 35) | 0.97 |
| EAEC *aggR* | 31.06 (N = 82) | 27.88 (N = 29) | 29.17 (N = 28) | 0.82 |
| EAEC *aaiC* | 17.80 (N = 47) | 25.96 (N = 27) | 22.92 (N = 22) | 0.18 |
| *Campylobacter jejuni/coli* | 14,39 (N = 38) | 11,54 (N = 12) | 13.54 (N = 13) | 0.77 |

%: percentage; N: total number of sample analysis; MS: moderately stunted; SS: severely stunted;

Comparisons between groups (controls and stunted children MS+SS) were determined using Pearson's χ2-test or Fisher's exact test, as appropriate. Only values with $p < 0.05$ could be considered to be statistically significant*.

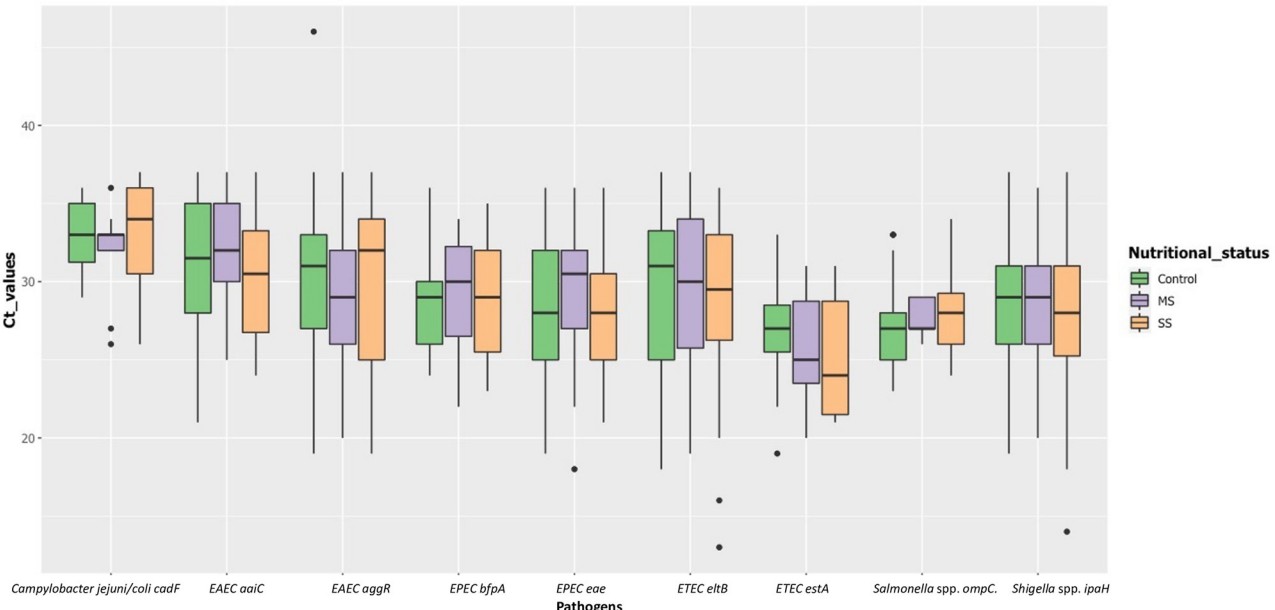

**Fig 3. Box plot showing C$_T$ values for pathogens targeted by real-time PCR among stunted (104 moderately stunted and 96 severely stunted) and controls (N = 264).** Boxes show the median (midline) and the 25$^{th}$ and 75$^{th}$ percentiles, and bars indicate the 10$^{th}$ and 90$^{th}$ percentiles. Only the positive subjects (C$_T$ values = <37) for the respective pathogen agent was considered.

## Hierarchical clustering of enteropathogens according to their virulence factors and nutritional the status of children

A hierarchical clustering based on a binary distance and Ward's agglomeration method allowed us to highlight 8 different clusters of pathogen detection profiles (Fig 4). A linear model was then applied to predict if HAZ correlated with the different virulence genes (*invA*, *ipaH*, *estla*, *eltB*, *aggR*, *aaiC*, *bfpA*, *eae* and *cadF*). The model showed a statistically not significant and weak proportion of variance ($R^2 = 0.03$, $F(9, 450) = 1.34$, $p = 0.214$, adj. $R^2 = 6.61e-03$). However, within this model, the effect of *aaiC* was statistically significant and negative (beta = -0.31, 95% CI [-0.58, -0.03], $t(450) = -2.19$, $p < .05$).

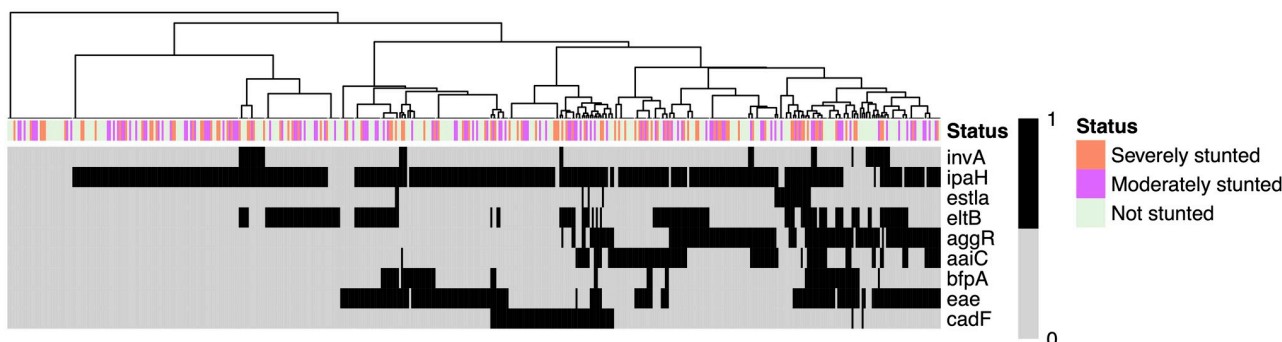

**Fig 4. Heatmap showing gene frequency detected by qPCR and height for age Z (HAZ) scores (x-axis: isolates; y-axis: targeted genes).** The dendrogram fitted on the isolates (i.e. the subjects) were computed based on an asymmetric binary distance coupled with a Ward agglomeration method based on the presence—black color—or absence of the genes—gray color. The clusters derived from the hierarchical clustering were not shown to be significantly associated to HAZ. No clustering was performed on the variables (targeted genes).

## *Shigella* spp. carriage

During the study, we decided to isolate *Shigella* spp. from the feces collected (N = 143 samples) by cultivating them on Hektoen and XLD media. Very few green or pink colonies grew respectively on Hektoen or XLD [from 41 feces samples] on which 1 representative was isolated for further study. Only two isolates (from HJRA178—HJRA198 samples) were positive (qPCR) and were identified by API20E as *Shigella flexneri* (ID scores of 69.1 and 82.4%). They were isolated from a severely (HJRA178) and a moderated stunted child (HJRA198), both female, showing no recent clinical sign of diarrhea.

The two isolates from HJRA178—HJRA198 samples were sequenced and their ANI and dDDH values based on full genome sequence regarding the closest relatives' strains were significantly higher than the recommended cut-off points for species boundary [27] to indicate that they belong to the genus *Shigella* and the species *flexneri* with the closest relative being *Shigella flexneri* ATCC 29903 (S6 Table). In addition, the phylogenomic tree constructed on the TYGS using FastME from the genome blast distance phylogeny (GBDP) [26,27] provided further evidence on the taxonomic position of these strains within the genus *Shigella* and the species *flexneri* (S2 Fig). These two isolates clustered with *Shigella flexneri* ATCC 29903. Genotypic differences between the two isolates were also found based on their MLST, wgMLST and cgMLST and virulence factors (S7 Table).

The two *Shigella* isolates from HJRA 178 and HJRA 198 stool samples were also referred to the French National Reference Center for Shigella at Institut Pasteur, Paris, France and were serotyped as *S. flexneri* 1b and *S. flexneri* 4v, respectively.

As the two *Shigella* isolates were isolated from feces of asymptomatic stunted children, we investigated whether the two isolates of *S. flexneri* were able to invade Hep2 cells by using the classical gentamicin protection assay [27]. After incubation and treatment with gentamicin the total number of Hep2 cell- internalized bacteria was calculated and compared to a wild-type (M90T) and an invasion-deficient (*mixD*) strain. The gentamicin assay was realized in duplicate. Invasion was defined as the total number of intracellular bacteria in cells (extracellular bacteria were killed by gentamicin, a cell-impermeable antibiotic). The HJRA178 isolate (*S. flexneri* 1b = 3109 in S3 Fig) displayed the same behavior than the invasion-deficient (*mixD*) mutant and the HJRA198 isolate (*S. flexneri* 4v = 3110 in S3 Fig) the same behavior than the wild-type (M90T) strain. This confirmed that one of the two strains had intact invasiveness properties while the other one did not. The HJRA178 isolate displayed the Congo red binding (Crb$^+$) phenotype associated the infectivity properties of *S. flexneri* whereas the HJRA198 isolate did not (Crb$^-$) confirming the data obtained in the gentamicin protection assay.

## Discussion

In this study, duodenal aspirates from stunted children (N = 109) were analyzed to characterize the bacterial population in the duodenum of stunted children and to investigate SIBO. Unfortunately, there is currently no single valid test for SIBO, and the accuracy of all current tests remains limited due to the failure of culture to be a gold standard. Even the value of $10^5$ CFU/ml has begun to be questioned by consensus and review of the literature. Some studies suggest normal subjects rarely exceed $10^3$ CFU/ml and that this should be the defining threshold for SIBO [32].

However, if we consider SIBO defined as greater than $10^5$ CFU/ml upper intestinal aspirate as assessed by both anaerobic and aerobic cultures, this study confirms the preliminary results obtained in the Afribiota study [15] showing a very high prevalence of SIBO in stunted children (85.3%) with 58.7% having more than $10^6$ bacteria/ml of duodenal fluid. Studies of children living in shantytowns in South America and Asia have detected SIBO at lower

prevalence; from 16% (Bangladesh) to 61% (São Paulo, Brazil) [13,33,34,35]. These high SIBO positivity rates in our study also reflect the fact that the cultivation approach is more sensitive than the breath tests utilized in other studies. The presence of excessive bacteria in the small intestine is typically associated with a malabsorptive syndrome occurring in the context of gut stasis syndromes [36]. It was also suggested that SIBO is associated with intestinal inflammation [13], especially in resource-poor communities where an extreme poverty and unhygienic conditions prevail.

The duodenum and proximal jejunum normally contain small numbers of bacteria ($< 10^4$ organisms per mL), usually lactobacilli, enterococci, streptococci (from the *Lactobacillales* order), gram-positive aerobes such as *Veillonella* (*Firmicutes*) or facultative anaerobes from *Moraxellaceae* and transiently non-oral genera from *Enterobacteriaceae* [37]. In this study, the most prevalent genera cultivated from the small intestine were *Streptococcus*, *Neisseria*, *Staphylococcus*, *Rothia*, *Haemophilus*, *Pantoea* and *Branhamella*. Those genera are generally oral taxa [38] and their presence in high numbers in duodenal fluids is uncommon. Our previous study on a relatively small number of stunted children in Antananarivo (Madagascar) and Bangui (CAR) showed a similar microbiota in both the stomach and duodenum of stunted children and a clear over-representation of pathobionts and of oropharyngeal species in the stools suggesting a decompartmentalization of the gastrointestinal tract [15]. We speculate that in these stunted children, gut maintenance declines and microbes can stray from traditional zones, negatively impacting intestinal homeostasis, host health and altered nutriment absorption [15].

In a recent study conducted on stunted undernourished children with enteropathy living in Bangladesh, the authors showed after esophagogastroduodenoscopy and duodenal biopsies a shared group of 14 taxa (not typically classified as enteropathogens) negatively correlated with linear growth and positively correlated with duodenal proteins involved in inflammatory responses [39]. The three most strongly correlated bacteria with the duodenal inflammatory markers were a *Veillonella* species, a *Streptococcus* species and *Rothia mucilaginosa*. Dysbiosis in the oral microbiota, and the significant increase of *Veillonella* sp. also positively correlated with elevated levels of inflammatory cytokines (IL-1ß, IFN-ɣ, TNF-α, IL-8) and immunoglobulin A in the saliva in patients with autoimmune liver disease [40] and in patients with inflammatory bowel disease (dominant genera: *Streptococcus*, *Prevotella*, *Neisseria*, *Haemophilus*, *Veillonella* and *Gemella* correlating to increased IL-1ß and lysozyme levels) [41]. The collected data emphasizes the possible role of the ectopic colonization in the upper part of the small intestine by some oral bacteria with pro- inflammatory features that could be tested in animal models with our isolated bacteria from the duodenum. Another explanation might be lack or the reduced concentration of gastric acid in the stunted children in LMICs (i.e. due to *Helicobacter pylori* contamination) which may lead to this bacterial overgrowth since gastric secretions is one of the normal defense mechanisms controlling the intestinal microflora [42]. However, the pH values measured in the stomach samples of our stunted children were low and there were no significant differences in a bivariate analysis for pH and SIBO (S4 Table).

This study also assessed the possible association of SIBO with potential risk factors among the participants (education, possible exposure to contaminants, hygiene, . . .). Although some studies on SIBO have revealed an association with some risk factors such as open sewer outside the home (OR, 4.78; 95% CI, 1.06 to 21.62) [13], the present study did not find any association between SIBO and the variables assessed. The absence of association could be due to the high prevalence of SIBO in this study (85.3% vs 16.7% and 13.5% for the studies referred to in 34 and 35).

The current research also aimed to assess by a molecular approach the carriage of intestinal pathogens in the feces, one of the important etiological causes which may be associated to EED

in the children population. Despite the disadvantages of molecular approaches, such as being unable to determine whether bacteria are still alive or metabolically active [43], they offer a high level of sensitivity. For example, it has been shown that systematic application of multiplex qPCR enhances from 18 to 30% the detection of bacteria, parasites, and viruses in stool samples [44,45]. The current study shows a high prevalence of bacterial enteropathogens, especially those categorized as "enteroinvasive" or causing mucosal disruption such as *Shigella* spp., ETEC, EPEC and EAEC. Unexpectedly, all these pathogens were detected at similar rate in MS/SS and in C (no statistical differences) and all the children showed no sign of severe diarrhea 15 days before the inclusion. Particularly, *Shigella* spp. was highly prevalent in the qPCR approach, also confirmed by a conventional PCR approach on a limited number of samples. To assert the presence of living *Shigella* spp. in the feces samples, isolation was performed at the end of the cross-sectional study directly from stool streaked on XLD and Hektoen plates and only two samples gave rise to two *S. flexneri* isolates confirmed by whole genome sequence analysis and serotyping. This very low recovery of *Shigella* isolates could be explained by the fact that no specific enrichment broth exists for *Shigella* (except Selenite-F [46]) or that there were no-metabolically active *Shigella* in the samples.

Subclinical infections (an infection in which symptoms are either absent or sufficiently mild to escape diagnosis [47]) or asymptomatic carriage (one who harbors pathogenic organisms without clinically recognizable symptoms and may infect others [48]) of intestinal pathogens seems common in low- and middle-income countries suffering from poor sanitary conditions. For instance, in Rwanda, high proportions of asymptomatic carriage (young healthy controls) were observed for adenovirus (50%), ETEC—based on *eltB* detection—(47%), EPEC—*eae*—(23%), *Campylobacter* (22%) and *Shigella*—*ipaH*—(17%) [49]. In children living in the Peruvian amazon, *Shigella* isolates (*S. flexneri* accounted for 67.1% of isolates) were also obtained by bacteriological cultures from 3.2% of surveillance stool cultures in the absence of diarrheal illness [50].

Asymptomatic carriage in young children has been attributed to several factors, including breastfeeding and maternal immunity, immunity gained from previous clinical or subclinical infections, and the intestinal microbiome acting as a gut barrier (which affects the likelihood that an enteric pathogen induces disease) [51]. In the case of *Shigella* it can be also interpreted as a prolonged pathogen excretion after illness (up to 17 months) [50] or a reinfection with the same strain from a very contaminated area with short- to medium term and serotype-specific immunity in children provided following clearance [52]. The carriage of a microorganism with limited pathogenicity could be also considered what could be the case for one of the two *Shigella* isolates (from the HJRA178 sample) as shown by the genomic analysis of the virulence factors and the inability to invade Hep2 cells. The limited pathogenicity could be explained by the loss of the virulence plasmid harboring the Mxi/Spa secretory apparatus encoded by two operons comprising about 25 genes. However, the other *Shigella* isolate (from the HJRA198 sample) was fully virulent and invasive suggesting different patterns of infection.

The MAL-ED Network Investigators [11] study showed an association between enteropathogens (mainly *Campylobacter* and EAEC) in non-diarrheal stools and reduced linear growth (length and weight). The high proportion of enteropathogens in this study, especially *Shigella* spp. but also ETEC and EAEC, in both populations (stunted and non-stunted), supports the fact that carriage of enteropathogens could not be only associated with the population of stunted children and could prevented us to propose a direct association with stunting. However, in our previous study conducted on a more restricted number of samples by 16S rDNA sequencing, members of *Escherichia coli*/*Shigella* and *Campylobacter* sp. were more prevalent in stunted children compared with non-stunted controls [15]. Our data suggest that, beside combatting poverty and improving diet, environmental sanitation, quality of water sources,

hygiene promotion and health education are key points to mitigate stunting and restore nutritional benefits even if a recent study provided evidence for the important role of timing of stunting on the recovery from the phenomenon rather than WASH practices [53].

## Limitations

This study has however some limitations such as a single run of qPCR amplification on individual samples. Moreover, the integration of qPCR amplification results for enteric parasites [43] and viral infections was not considered either.

Regarding the duodenal sampling technique via nasogastric tube, we ensured the first ml of aspiration from both stomach and duodenum were discarded in order to flush out possible contaminating bacteria and carry-over from the more proximal compartments.

This study was not intended to directly measure evidence of EED but two investigate two major–possibly combined—etiologies, which may account for EED: SIBO and exposure to enteropathogens.

## Supporting information

**S1 Fig. Distributions of the CFU in duodenal aspirates.** Distributions of the CFU in duodenal aspirates with SIBO matching with feces not contaminated by *Campylobacter* spp. [0 (in blue)]; and in duodenal aspirates with SIBO matching with feces *contaminated* by *Campylobacter* spp. [1 (in yellow)].
(TIF)

**S2 Fig. Phylogenomic tree.** The phylogenomic tree includes the two *Shigella* isolates from HJRA178—HJRA198 samples constructed on the Type Strain Genome Server (TYGS) (at https://tygs.dsmz.de) using FastME from the genome blast distance phylogeny (GBDP). [26,27].
(TIF)

**S3 Fig. Ratio of intracellular bacteria per input for the two *Shigella flexneri* isolates (from HJRA178 = 3109 and HJRA198 = 3110 samples) and for the mutant *mxiD* (non-invasive) and the strain M90T (invasive).**
(TIF)

**S1 Table. Lists all data (risk factors) collected and used in univariate analysis with SIBO.**
(XLSX)

**S2 Table. Results for univariate analysis and multivariate analysis.**
(XLSX)

**S3 Table. Bivariate analysis for stomach pH and SIBO.**
(PDF)

**S4 Table. SIBO and qPCR results.**
(XLSX)

**S5 Table. Correlation between the pathogen presence in stools with the measure of CFU/ml grown from duodenal aspirates in aerobic and anaerobic culture conditions using the Wilcoxon's rank sum test.**
(XLSX)

**S6 Table. ANI and dDDH values based on genome sequences of the two isolated strains from HJRA178–HJRA198 samples and the closest relatives' strains.**
(XLSX)

**S7 Table. Virulence factors of the two isolated strains from HJRA178–HJRA198 samples.**
(XLSX)

**S1 Text. List pf all AFRIBIOTA Investigators.**
(PDF)

## Acknowledgments

We wish to thank all participating families, the AFRIBIOTA Consortium, the participating hospitals in Antananarivo, as well as the Institut Pasteur, the Institut Pasteur de Madagascar, and members of the scientific advisory board for their continuous support; Prof. Jean-Louis Demarquez for training sessions to teach the local health professionals the methods used for duodenal aspirations; Aurélie Etienne for precious help with the clinical procedures and first aspirations performed; the field workers Tseheno Harisoa and Rado Andrianantenaina, as well as all implicated community health workers, for countless hours spent in the field; the Centre de Recherche Translationelle and the Direction Internationale of the Institut Pasteur, and especially Paméla Palvadeau, Jane Lynda Deuve, Marc Rovatiana Ranarijesy, Kanto Liantsoa Razanakolona, Cécile Artaud, Nathalie Jolly, Sophie Jarrijon, Mamy Ratsialonina, Jean-François Damaras, Marie-Noelle Ungeheuer, and Laurence Arowas for precious help in setting up and steering the AFRIBIOTA project and managing the funds and the biobank. We would like to thank the staff of the "Plateforme de Microbiologie Mutualisée (P2M)" at Institut Pasteur Paris where the whole genome sequencing was performed, Sophie Lefevre from the Centre National de Référence des *Escherichia coli*, *Shigella* et *Salmonella* in Paris, France for *Shigella* serotyping, and Claude Parsot and Alexandre Grassart for the interesting discussions on *Shigella*.

Thanks are also due to Daniel Falush and Khashayar Shahin for carefully reading and commenting the manuscript.

## Author Contributions

**Conceptualization:** Jean-Marc Collard, Vincent Guillemot, Pascale Vonaesch, Philippe J. Sansonetti.

**Data curation:** Jean-Marc Collard, Maheninasy Rakotondrainipiana, Prisca Andriantsalama, Ravaka Randriamparany, Rindra Vatosoa Randremanana, Vincent Guillemot, Pascale Vonaesch.

**Formal analysis:** Jean-Marc Collard, Lova Andrianonimiadana, Azimdine Habib, M. A. N. Rabenandrasana, François-Xavier Weill, Nathalie Sauvonnet, Vincent Guillemot, Pascale Vonaesch, Philippe J. Sansonetti.

**Funding acquisition:** Pascale Vonaesch, Philippe J. Sansonetti.

**Investigation:** Jean-Marc Collard, Maheninasy Rakotondrainipiana, Prisca Andriantsalama, Ravaka Randriamparany, Rindra Vatosoa Randremanana, Pascale Vonaesch, Philippe J. Sansonetti.

**Methodology:** Jean-Marc Collard, Lova Andrianonimiadana, Azimdine Habib, M. A. N. Rabenandrasana, François-Xavier Weill, Nathalie Sauvonnet, Vincent Guillemot, Pascale Vonaesch, Philippe J. Sansonetti.

**Project administration:** Jean-Marc Collard, Pascale Vonaesch, Philippe J. Sansonetti.

**Resources:** Jean-Marc Collard, Pascale Vonaesch, Philippe J. Sansonetti.

**Software:** Lova Andrianonimiadana, Azimdine Habib, Maheninasy Rakotondrainipiana, Prisca Andriantsalama, Ravaka Randriamparany, M. A. N. Rabenandrasana, Rindra Vatosoa Randremanana, Vincent Guillemot, Pascale Vonaesch.

**Supervision:** Jean-Marc Collard, Pascale Vonaesch, Philippe J. Sansonetti.

**Validation:** Jean-Marc Collard, Lova Andrianonimiadana, Azimdine Habib, M. A. N. Rabenandrasana, François-Xavier Weill, Nathalie Sauvonnet, Vincent Guillemot, Pascale Vonaesch, Philippe J. Sansonetti.

**Visualization:** Jean-Marc Collard, Lova Andrianonimiadana, Azimdine Habib, Maheninasy Rakotondrainipiana, Prisca Andriantsalama, Ravaka Randriamparany, M. A. N. Rabenandrasana, Nathalie Sauvonnet, Rindra Vatosoa Randremanana, Vincent Guillemot, Pascale Vonaesch.

**Writing – original draft:** Jean-Marc Collard, Azimdine Habib.

**Writing – review & editing:** Jean-Marc Collard, Lova Andrianonimiadana, Azimdine Habib, Maheninasy Rakotondrainipiana, Prisca Andriantsalama, Ravaka Randriamparany, M. A. N. Rabenandrasana, François-Xavier Weill, Nathalie Sauvonnet, Rindra Vatosoa Randremanana, Vincent Guillemot, Pascale Vonaesch, Philippe J. Sansonetti.

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
