## [Decision Letter · Decision Letter 0]

11 Nov 2021

Dear Dr Collard,

Thank you very much for submitting your manuscript "High prevalence of small intestine bacteria overgrowth and asymptomatic carriage of enteric pathogens in stunted children in Antananarivo, Madagascar." for consideration at PLOS Neglected Tropical Diseases. As with all papers reviewed by the journal, your manuscript was reviewed by members of the editorial board and by several independent reviewers. In light of the reviews (below this email), we would like to invite the resubmission of a significantly-revised version that takes into account the reviewers' comments. 

We cannot make any decision about publication until we have seen the revised manuscript and your response to the reviewers' comments. Your revised manuscript is also likely to be sent to reviewers for further evaluation.

Sincerely,

Andrew S. Azman

Deputy Editor

Reviewer's Responses to Questions

**Key Review Criteria Required for Acceptance?**

**Methods**

-Are the objectives of the study clearly articulated with a clear testable hypothesis stated?

-Is the study design appropriate to address the stated objectives?

-Is the population clearly described and appropriate for the hypothesis being tested?

-Is the sample size sufficient to ensure adequate power to address the hypothesis being tested?

-Were correct statistical analysis used to support conclusions?

-Are there concerns about ethical or regulatory requirements being met?

Reviewer #1: The methods are well described and are appropriate.

Reviewer #2: -Enrollment criteria need to be reviewed in detail. “other severe diseases” is vague.

-Were all children truly admitted to the hospital as stated in line 119? If so, reason for admission needs to be detailed.

-It seems that duodenal sampling technique via nasogastric tube could introduce nasopharyngeal flora into the sample thus contaminating it and skewing results for an over-representation of oral taxa. Authors should address this in methods and/or the limitations section of the discussion. 

-Data collection on co-variates is unclear. A complete list of what was assessed in univariate analysis needs to be provided with details (ie. Who assessed for dental carries). If extensive then perhaps a correction for multiple comparisons is appropriate. 

-Can authors cite a reference for using a CT cutoff of 37? This seems higher than most qPCR analyses. How was this value chosen?

-eae, as authors state, could represent EHEC as well as EPEC. When authors use “either target” this may over-represent EPEC with some of these isolates being EHEC.

**Results**

-Does the analysis presented match the analysis plan?

-Are the results clearly and completely presented?

-Are the figures (Tables, Images) of sufficient quality for clarity?

Reviewer #1: Yes

Reviewer #2: -Would add the number of duodenal aspirates to line 287.

-Line 289 – 292 “The duodenum and proximal jejunum…” this is more fitting in the background. Not in the results. 

-A better description of how colonies and morphotypes were selected for MALDI-TOF is needed. Were a random number chosen from each sample? Were all isolates cultures sent for MADLI-TOF. This has major implications on the results so this needs to be clear. The description of this should be in the methods, not results.

-Line 306 – 208 suggests no risk factors for SIBO were identified however in Table 3 handwashing is significant. Please clarify. Is table 3 univariate only (if so, suggest changing to multivariate final model). 

-Line 316 states no significant difference was found between pathogen carriage amongst stunting groups. However, in Table 4 Salmonella and Shigella are statistically significant. Please clarify. 

-Line 319 reports mean, median, and range of CT values. Is this for all pathogens and targets? 

-Line 320 states no difference in CT values was noted between groups, CT values for which pathogen? All pathogens? Please clarify.

-Line 322-323: Meaning isn’t clear. 1/3 of the children had one of the 3 tested E. coli pathotypes?

-It is unclear what exactly the cluster dendrogram of Fig 4 was clustered on. Details need to be provided in the methods section.

-The difference in positivity between conventional and qPCR needs to be explained. Do you this qPCR has false positives or does traditional PCR miss cases? I suggest the authors make an argument for a single method and limit reporting to that method.

-Analysis on Shigella carriage seems unrelated to the SIBO story. While interesting, I would suggest removing it and submitting as a separate report. 

-Table 3: Is this univariate or multivariate. Title should be more descriptive. 

-Figure 1: The abbreviation SR is not defined in the figure legend. 

-Figure 2: No microbiome analysis was conducted and thus the units on the Y axis of “relative abundance” do not make sense. Also, the range of 0-500 does not make sense if this is relative abundance. Please clarify. 

-Figure 4: Legend needs to be expanded to describe what variables were clustered on here.

**Conclusions**

-Are the conclusions supported by the data presented?

-Are the limitations of analysis clearly described?

-Do the authors discuss how these data can be helpful to advance our understanding of the topic under study?

-Is public health relevance addressed?

Reviewer #1: yes

Reviewer #2: -Authors state they “precisely characterize the bacterial population in the duodenum”. This is an overstatement as they utilized outdated culture techniques rather than molecular means of quantifying the SIBO dysbiois. This should be listed as a limitation and this sentence revised. 

-Would discuss differences in positivity rates between this study and others in the literature. Perhaps culture of duodenal aspirate is more sensitivity than the breath testing utilized in the other studies discussed. 

-Authors suggest that SIBO causes inflammation, villus blunting, and decreased crypt-to-villous ratio. The relation of SIBO to histology needs to be cited. Further, stating the relationship is causal is an overstatement. “Associated” is more appropriate. 

-Line 399 – 400 “Those genera are generally…”: Needs citation.

-Line 406 – 407 “In these stunted children…”: Sentence is purely speculative and none of this was shown in the cited reference. 

-Line 413: This is misleading. Strep, Veillonella, and Rothia are often found in the small intestine as well as the oral cavity.

-The discussion of URI and children swallowing oral flora is purely speculative and without evidence. This should be removed. 

-Again, suggest revising all instances where authors state a causal relationship between pathogen carriage and SIBO/EED.

-Limitations section should be moved to the discussion. Sampling technique and possible contamination of oral taxa on the nasogastric tube as well as lack of molecular testing on aspirates (i.e. culture techniques) should be added as limitati

**Editorial and Data Presentation Modifications?**

Reviewer #1: The results section describing the two shigella isolates is too wordy - this can be shortened to very few lines.

Reviewer #2: -There are many typos throughout this manuscript which are irksome to the reviewer

-English and grammar are incorrect throughout the manuscript. This obscures meaning in places.

**Summary and General Comments**

Reviewer #1: This is a well written article that reports high rates of small intestinal overgrowth in stunted, but not acutely ill children aged 2 to 5 years. The report is consistent with earlier studies in the literature so in this way, the data are mostly confirmatory, but the finding of oral bacteria in the upper small intestine is interesting. The finding of such a high prevalence of Shigella in fecal samples of all groups is also similar to other studies, except this is even higher. It is a bit unusual to find such a low number of these could be cultured. Shigella culture is not sensitive because of lack of enrichment broth, but it is strange to only recover Shigella from two samples. 

Some specific suggestions:

Line 353-380 The description of the two Shigella isolates is too wordy. The data is important, but it could be said using many fewer lines. 

Regarding the discussion section. The authors suggest the intestinal bacteria are coming from the oral cavity and are perhaps protected from stomach acid by oral or nasal mucus. Another explanation might be lack of gastric acid in the stunted children and this may also be associated with Helicobacter infection. See Gracey M, et al. The stomach in malnutrition. Arch Dis Child 1977;52:325-7 and Windle HJ, et al Childhood Helicobacter pylori infection and growth impairment in developing countries: a vicious cycle? Pediatrics 2007;119:e754-9. The authors have information on the pH of the stomach (especially if stimulated) or information on rates of infection with Helicobacter in this population?

If the authors are proposing the intestinal bacteria originate from the oral cavity, what are the suggestions for future studies to explore this. Also, how did the intubation method exclude the potential that the fluid obtained was not contaminated by the nasal bacteria as it was passing through the nasal cavity?

The section on limitations is too brief. Some additional limitations include: 

1) the study did not directly measure evidence of EE, but they assumed this.

2) the finding of the bacteria in the small intestine and stool shows an association with malnutrition, but this does not prove cause and effect. While came first? 

3) the study did not attempt to correlate oral and intestinal bacteria.

Table 4 should clarify that these are fecal samples

Reviewer #2: OVERALL: 

The authors present an interesting analysis of children in Madagascar investigating the dysbiotic microbiome associated with SIBO using culture methods. They attempt to determine risk factors for SIBO as well as describe enteric pathogen carriage. They report 2 Shigella isolates which were sequenced and further investigated. This analysis has significant overlap with reference 14 and likely utilizes some of the same samples although this is unclear. Major critiques center around unrelated analyses being reported and unclear methodology. The introduction and discussion sections are also highly speculative. 

Further, it seems there is a missed opportunity here to investigate the relationship of enteric pathogen carriage to SIBO. I would suggest correlating pathogen presence or burden (CT value) with the continuous measure of CFU/ml of growth on duodenal aspirate culture. 

ABSTRACT:

-The preferred term is Environmental Enteric Dysfunction (EED), not Environmental Enteric Disease. EE generally stands for Environmental Enteropathy, a term that has gone out of favor in recent years.

-While EED has certainly been associated with enteric pathogen carriage, mechanisms of pathogenesis have yet to be fully explained. Throughout the manuscript, authors should refrain from stating that EED is caused by pathogen carriage.

-EED is a complex syndrome, part of which includes an upper intestinal dysbiosis but to say EED causes a dysbiosis is misleading and unproven. 

-Lines 40 – 44 “The presence of those commonly…”: This sentence is purely speculative without evidence. Would remove such speculation.

INTRODUCTION:

-Overall, much of the introduction is author opinion/speculation which is inappropriate in scientific writing. I suggest the authors focus on reviewing the relevant literature (cited appropriately and correctly) to suggest the current study is needed. 

-Line 75: Adults in LMICs also have evidence of EED although this is less studied.

-Line 76: The term “developing world” is no longer preferred. Recommend using the World Bank classification of low- and middle-income countries.

-Lines 86 – 89 “The second, non-exclusive, etiology…” This entire sentence is unproven speculation. Citations are inappropriately used and neither discusses oral taxa, taxa associated with SIBO, or SIBO histology. 

-Authors need to be clear in the introduction as to how this manuscript differs from reference 14 and adds something novel to the literature.

PLOS authors have the option to publish the peer review history of their article (what does this mean?). If published, this will include your full peer review and any attached files.

Reviewer #1: No

Reviewer #2: No
---

## [Decision Letter · Decision Letter 1]

23 Feb 2022

Dear Dr Collard,

Thank you very much for submitting your manuscript "High prevalence of small intestine bacteria overgrowth and asymptomatic carriage of enteric pathogens in stunted children in Antananarivo, Madagascar." for consideration at PLOS Neglected Tropical Diseases. As with all papers reviewed by the journal, your manuscript was reviewed by members of the editorial board and by several independent reviewers. The reviewers appreciated the attention to an important topic. Based on the reviews, we are likely to accept this manuscript for publication, providing that you modify the manuscript according to the review recommendations. 

Sincerely,

Andrew S. Azman

Deputy Editor

Reviewer's Responses to Questions

**Key Review Criteria Required for Acceptance?**

**Methods**

-Are the objectives of the study clearly articulated with a clear testable hypothesis stated?

-Is the study design appropriate to address the stated objectives?

-Is the population clearly described and appropriate for the hypothesis being tested?

-Is the sample size sufficient to ensure adequate power to address the hypothesis being tested?

-Were correct statistical analysis used to support conclusions?

-Are there concerns about ethical or regulatory requirements being met?

Reviewer #1: yes

Reviewer #2: -If the samples from ref 15 (153 fecal and 12 duodenal) were included in the data set used here then this needs to be explicitly stated so readers understand the duplication of published data. If there is overlap, I would add a sentence stating that “153 fecal sample and 12 duodenal samples utilized included in this analysis have been previously analyzed and published on”(with appropriate reference).

-It is mentioned in the methods that time between defecation and freeing was tracked. I would report the average either in the methods or in the first paragraph of the results.

-When authors state the first ml of aspirate was discarded for “flashing” out of possible contamination, do you mean ‘flushing’?

**Results**

-Does the analysis presented match the analysis plan?

-Are the results clearly and completely presented?

-Are the figures (Tables, Images) of sufficient quality for clarity?

Reviewer #1: Yes

Reviewer #2: -All presented data and supplementary information (i.e. Supplementary tables S1 and S2) need to be in English as this is an English journal. Please translate. 

-The finding of SIBO’s association with Campylobacter potentially quite significant. Neither in the manuscript nor in table S4 is directionality shown. Would add average CFU/ml in Campy positive and negative subjects with SD to the body of the manuscript (line 347) so readers can understand this relationship.

**Conclusions**

-Are the conclusions supported by the data presented?

-Are the limitations of analysis clearly described?

-Do the authors discuss how these data can be helpful to advance our understanding of the topic under study?

-Is public health relevance addressed?

Reviewer #1: Yes

Reviewer #2: -Line 435 – 437: This is the first mention of pH values being the same between stunted and control children. This, and the reference to S7 Table, should be moved to the results section as new results should not be first presented in the discussion.

**Editorial and Data Presentation Modifications?**

Reviewer #1: the authors carefully responded to the reviewer's comments.

Reviewer #2: -Issues with grammar and typos remain but much improved. 

-Abbreviations should be defined at their first use (see line 114: SARI/ILI).

**Summary and General Comments**

Reviewer #1: This study includes a substantial amount of data regarding intestinal overgrowth and makes some interesting observations about relationship with oral organisms. The stool microbiology is interesting, but is largely confirmatory. It is interesting that the apparent "pathogens" did not significantly relate to the degree of malnutrition.

Reviewer #2: Authors have adequately addressed concerns in the first manuscript with a few minor suggestions remaining. I continue to think that this is a unique and important analysis with potentially large impact on the field of environmental enteric dysfunction and malnutrition.

PLOS authors have the option to publish the peer review history of their article (what does this mean?). If published, this will include your full peer review and any attached files.

Reviewer #1: No

Reviewer #2: No

Figure Files:

Data Requirements:

Reproducibility:

References

---

## [Editor Report · Decision Letter 2]

1 Apr 2022

Dear Dr Collard,

We are pleased to inform you that your manuscript 'High prevalence of small intestine bacteria overgrowth and asymptomatic carriage of enteric pathogens in stunted children in Antananarivo, Madagascar.' has been provisionally accepted for publication in PLOS Neglected Tropical Diseases.

Best regards,

Andrew S. Azman

Deputy Editor

---

## [Editor Report · Acceptance letter]

2 May 2022

Dear Dr Collard,

We are delighted to inform you that your manuscript, "High prevalence of small intestine bacteria overgrowth and asymptomatic carriage of enteric pathogens in stunted children in Antananarivo, Madagascar.," has been formally accepted for publication in PLOS Neglected Tropical Diseases.

Best regards,

Shaden Kamhawi

co-Editor-in-Chief

Paul Brindley

co-Editor-in-Chief
